**PLOS** NEGLECTED TROPICAL DISEASES

# Community-Based Mycetoma Surveillance in Uganda: Identifying Knowledge Gaps and Training of Community Health Workers to Improve Case Detection

**Winnie Kibone** [1,2]*, **Andrew Weil Semulimi**[3], **Richard Kwizera**[3,4,5], **Felix Bongomin**[2]*

**1** Department of Medicine, School of Medicine, Makerere University, Kampala, Uganda, **2** Department of Medical Microbiology and Immunology, Faculty of Medicine, Gulu University, Gulu, Uganda, **3** Makerere University Lung Institute, College of Health Sciences, Makerere University, Kampala, Uganda, **4** Department of Research, Infectious Diseases Institute, College of Health Sciences, Makerere University, Kampala, Uganda, **5** Departme1nt of Medical Microbiology, School of Biomedical Sciences, College of Health Sciences, Makerere University, Kampala, Uganda

\* kibonewinnie@gmail.com (WK); drbongomin@gmail.com (FB)

**Data Availability Statement:** All relevant data are within the paper and its Supporting Information files.

## Abstract

### Background

Mycetoma is an uncommon and neglected tropical disease in Uganda. We aimed to assess baseline knowledge and provide community health workers (CHWs) in Northern Uganda with knowledge to identify and refer presumptive mycetoma cases.

### Methodology

Between March and August 2023, we conducted a concurrent triangulation mixed methods study among CHWs in Gulu and Pader districts on mycetoma. We conducted a 1 day in-person training on mycetoma. Quantitative data were collected before (pretest), immediately (immediate posttest) and six months (6-month posttest) after the training and results compared using paired sample t test or one-way ANOVA. Qualitative data were collected using four focused group discussions, audio recorded, and analyzed using thematic content analysis.

### Principal findings

Forty-five CHWs were enrolled, mostly male (66.7%, n = 30), with a median age of 36 years (IQR 29 43). Out of a total score of 18, the baseline mean knowledge score was 7±2.42, improving to 11±1.99 immediately posttest (p<0.001), and 10±2.35 at 6 months (p<0.001), without additional training. Significant knowledge improvements at 6 months were observed among female participants (p = 0.004), those aged 30 40 years (p = 0.031) or 40+ years (p = 0.035), and those with secondary education (p = 0.007). Over 6 months, CHWs screened 2,773 adults, identifying and referring 30 presumptive mycetoma cases. Qualitative findings revealed challenges and barriers to early identification and referral of mycetoma

**Funding:** This study was funded by the Royal Society of Tropical Medicine and Hygiene (RSTMH) through the Early Career Grant, awarded to WK. The funders had no role in study design, data collection and analysis, decision to publish, or preparation of the manuscript.

**Competing interests:** The authors have declared that no competing interests exist.

presumptive cases including limited knowledge, stigma, myths, lack of an indigenous name for mycetoma, delayed decision making, and transportation barriers.

## Conclusions

This study highlights a significant knowledge gap among CHWs about mycetoma, with substantial improvement following training. The identification of presumptive cases by CHWs reflects their potential in community-based surveillance, emphasizing the need to integrate well-trained CHW to lead efforts for mycetoma surveillance and capacity building to enhance health outcomes in Uganda.

## Author summary

Mycetoma is a rare neglected tropical disease that affects the skin and the tissues beneath skin, causing severe health issues if not diagnosed and treated early. It disproportionately affects people in rural and impoverished communities in sub-Saharan Africa, including Uganda. Mycetoma often affects mostly young adults in their productive years especially those participating in agricultural activities. In our study, we trained community health workers (CHWs) in Northern Uganda for one day to improve their knowledge and skills in identifying and referring cases of mycetoma. Our results showed that the CHWs had significant knowledge improvement that persisted for at least six months without more training. Furthermore, they were able to identify 30 possible cases of mycetoma after assessing nearly 3,000 people in the communities. Furthermore, we found challenges and barriers to early detection of this disease such as limited initial knowledge, stigma, myths about the disease, and practical issues like transportation difficulties and long distances to health facilities. Therefore, our study shows that by empowering local health workers, we can improve early detection and treatment of mycetoma, ultimately enhancing health in affected regions.

## Introduction

Mycetoma, a chronic and debilitating disease, is caused by fungi (eumycetoma) or slow-growing bacteria (actinomycetoma). The most prevalent causative agents include *Madurella mycetomatis*, *Actinomadura madurae*, *Streptomyces somaliensis*, *Actinomadura pelletieri*, *Nocardia brasiliensis*, and *Nocardia asteroids [1,2]*. Mycetoma affects skin, subcutaneous tissue and bone, disproportionately impacting people in rural and impoverished communities in sub-Saharan Africa, including Uganda [3,4]. Mycetoma is associated with several complications such as severe deformities and disabilities that result in social exclusion, stigma and loss of economic productivity by affected persons and their families [5]. Mycetoma is prevalent in tropical and subtropical regions also known as "mycetoma belt" where a significant number of people walk barefooted [6]. Being an implantation mycosis, mycetoma often affects mostly young adults in their productive years especially those participating in agricultural activities [7].

In 2016, the World Health Organization (WHO) designated mycetoma as a neglected tropical disease, however, the true burden of mycetoma, including in regions with high endemicity, remains largely unknown. A comprehensive review conducted in 2020 by Emery and Denning identified 19,494 mycetoma cases across 102 countries from 1876 to 2019, averaging 135 cases

per year, with Sudan reporting the highest number, with approximately 10,608 cases [8]. However, these figures likely underestimate the true global burden due to underreporting and diagnostic challenges in endemic regions [9,10]. In Uganda, about 4,000 people are estimated to have mycetoma, with an average prevalence of 8.32 cases per 100,000 persons per decade [11]. In Uganda, mycetoma is endemic in several regions, however, the true disease burden is underestimated due to a lack of surveillance systems and inadequate diagnostic tools resulting in under-detection and underreporting of mycetoma cases.

The management of mycetoma requires timely referral, prompt and accurate diagnosis, followed by appropriate treatment, which often involves a combination of surgery and long-term antimicrobial therapy [12]. However, clinical case detection is hampered by the asymptomatic or minimally symptomatic nature of the disease, slow disease progression as well as myths and misconceptions on mycetoma which prompts patients to seek alternative forms of medication [13–15]. The index of clinical suspicion is also still low [11].

Community health workers (CHWs) also known as village health teams (VHTs) are the lowest cadres in the Ugandan health system. They have emerged as important contributors to healthcare delivery in Uganda, particularly in rural and underserved areas, where they play essential roles in addressing various illnesses such as HIV, malaria, tr*ypanosomiasis*, and childhood immunizable diseases through community outreach, education, surveillance, and early detection and referral [16]. As such, CHWs have contributed to the community-driven and self-sustainability of the health system. Despite their effectiveness in addressing multiple health challenges, CHWs have not previously been involved in identifying and addressing the burden of mycetoma. However, a recent publication by the Mycetoma working group emphasizes the importance of training CHWs as a form of building capacity and awareness to ensure early case detection, improve the cure rate and avoid unnecessary amputations [17]. In this context, CHWs would play a critical role in mycetoma control as the first point of contact between patients and the health system in Uganda [18,19]. However, the inadequacy in knowledge of mycetoma among CHWs poses a significant challenge in the detection and control of mycetoma in Uganda. As such, addressing this knowledge gap through training of CHWs can significantly improve mycetoma control in Uganda and contribute to the overall effort to eliminate neglected tropical diseases in sub-Saharan Africa to achieve universal health coverage and health equity. Leveraging existing community networks and trust, CHWs have the potential to bridge the gap between formal healthcare systems and remote populations, thereby enhancing access to healthcare and reducing diagnostic delays.

Therefore, in the present study, we aimed to identify knowledge gaps and evaluate the impact of training CHWs on mycetoma case detection in Northern Uganda. We also aimed to identify the barriers and facilitators to early identification and referral of suspected mycetoma cases among CHWs. Findings from this will inform the development of targeted training programs that address the identified gaps and improve the quality of care provided to mycetoma patients.

## Methods

### Ethics statement

The study was approved by the Mulago Hospital Research and Ethics Committee (MHREC-2406). We obtained administrative clearance from the offices of District Health Officers in Gulu and Pader districts. All participants signed a consent form to provide written informed consent before being enrolled in the study.

## Study design and settings

This was a concurrent triangulation mixed-methods study among CHWs to assess barriers and facilitators to early detection and referral of cases of mycetoma in Pader and Gulu districts in Northern Uganda. This approach was chosen to enable the simultaneous collection and comparison of qualitative and quantitative data [20]. Registered CHWs were recruited through the offices of the District Health Officers (DHO) in Gulu and Pader districts. The study was conducted between March and August 2023.

We leveraged the ongoing partnership between the Northern Uganda Medical Mission (NUMEM) and the Uganda Ministry of Health, which focuses on developing and expanding community health workers (CHWs) training. This collaboration has provided training to over 300 CHWs in all sub-counties in the mentioned districts. Each sub-county is comprised of several villages, with most villages having at least one CHW. These CHWs undergo annual reassessment of skill and didactic competencies. Upon achieving consistent competency scores of >85% for two or more years, CHWs can opt to become advanced (lead) CHWs, subsequently serving as trainers for new CHWs to ensure the sustainability of the CHW program.

## Study population

We included CHWs aged 18 years or older, formally registered by the office of the DHO, both male and female, who provided written informed consent to participate in the study. For this pilot study, we purposely selected 45 CHWs operating within the catchment areas of Awach Health Centre IV in Gulu and Pajule Health Centre IV in Pader district, Uganda.

## Data collection

**Pre-training assessment.** Before the training, we conducted a formative, quantitative and qualitative baseline assessment of the knowledge of CHWs about mycetoma.

Quantitative data was collected using a semi-structured paper-printed questionnaire that consisted of questions on demographic characteristics, knowledge on various aspects of mycetoma and pictorial identification of mycetoma and mimickers of mycetoma. The questionnaire was administered over 30 minutes in both English and *Acholi* by bilingual research assistants supervised by WK and FB (S1 Data). The questionnaire was developed based on fungal diseases books and briefings from the WHO and Global Action for Fungal Infection (GAFFI) (https://www.gaffi.org/wp-content/uploads/Mycetoma-briefing-paper-final-September-2018.pdf). The completed questionnaires were reviewed separately by two investigators (WK and FB), who then provided the knowledge scores. Disagreements were resolved through a consensus discussion.

Qualitative data were collected, while examining knowledge and awareness of CHWs on mycetoma. This included barriers and facilitators to early diagnosis and referral of suspected cases of mycetoma, as well as and training needs, in order to support and enhance early detection and referral of suspected cases of mycetoma. (S2 Data).

Qualitative data were collected on the knowledge and awareness of CHWs on mycetoma. This included barriers and facilitators to early identification and referral of suspected cases of mycetoma, as well as training needs and support to enhance early detection and referral.

A total of 4 focused group discussions (FGDs), 2 in each district, each consisting of 10–12 CHWs were conducted by a qualified social scientist (BSc, MPH) with vast experience in qualitative research. All FGDs were audio-recorded and filed notes were taken. During the FGDs, the participants were seated around a round table, with the moderator seated among them. At the start, the moderator shared with participants the objectives of the study. Each FGD lasted 1 hour.

Definition: A suspected case of mycetoma refers to an individual presenting with clinical symptoms that align with the characteristic features of the disease including painless swelling, draining sinuses, and the presence of grains.

**Intervention.** CHWs underwent a 1-day training covering aspects of mycetoma including, definition, causes, clinical presentation, diagnosis, management and mimics of mycetoma, and the role of CHWs in identification and referral of suspected mycetoma cases. The training curriculum, developed using existing materials from Leading International Fungal Education (LIFE) (https://life-worldwide.org/life-education-slide-sets-video-presentations-and-reading-materials), and WHO training guides, was delivered by two experienced clinical mycologists using a combination of PowerPoint presentations, tutorials, pictorials, and group discussions.

**Post-training evaluation.** Immediately after the training and at 6 months post-training, participants filled a semi-structured paper-printed questionnaire identical to the pre-test tool. The questionnaire was completed over a 30-minute period, and pre- and post-test questionnaires were independently reviewed by the investigators (WK and FB). Disagreements were resolved through consensus discussions. Additionally, a data extraction sheet was provided to each CHW for notification and referral of individuals with suspected mycetoma. The data extraction sheet included details on the age, sex, social and occupational information of the affected individual, the affected part of the body, symptoms, physical findings, and referral information. The CHWs were trained on how to fill the sheet.

## Data analysis

Quantitative data were analyzed using GraphPad Prism 8.03. and STATA version 17.0. Baseline characteristics were summarized using univariate descriptive analysis as mean and standard deviation for numerical variables and frequency and percent for categorical variables. Knowledge scores were calculated out of a total score of 18, and bivariate comparisons were made using paired sample t-test or one-way Analysis of Variance (ANOVA) as appropriate. A $p<0.05$ was considered statistically significant. The STROBE checklist was used, (S3 Data).

Qualitative data were transcribed verbatim and translated from *Acholi* into English. Our social scientist assigned codes to relevant segments of the text, and similar or related codes aggregated to form themes. We used thematic content analysis to analyze the qualitative data, which involved coding and identifying and categorizing themes that emerged from the data into themes and sub-themes. Qualitative data were reported according to consolidated criteria for reporting qualitative studies (COREQ) [21], (S4 Data).

## Results

### Baseline characteristics

We included 45 CHWs (30 males and 15 females) with a median age of 36 years (IQR = 29–43). Just over half of the CHWs were from Gulu (n = 23, 51.1%) and 35 (77.8%) had attained a secondary school level of education. One-third (n = 15, 34.1%) of the CHWs had more than 10 years of work experience as CHWs. However, only 9 (20.0%) had prior awareness of mycetoma, and only 1 (2.2%) had received training on mycetoma. Fourteen (31.1%) CHWs reported encountering a patient with suspected mycetoma within the past six months (**Table 1**). Overall, 38 (84.4%) CHWs reported mycetoma is curable.

**Table 1. Baseline characteristics of the study participants (CHWs).**

| Variable | Frequency | Percentage |
|---|---|---|
| **Sex** | | |
| Male | 30 | 66.7% |
| Female | 15 | 33.3% |
| **Age median (interquartile range)** | 36 (29–43) | 100% |
| <30 | 12 | 26.7% |
| 30–40 | 18 | 40.0% |
| >40 | 15 | 33.3% |
| **District of practice** | | |
| Gulu | 23 | 51.1% |
| Pader | 22 | 48.9% |
| **Highest level of education** | | |
| Primary | 3 | 6.67% |
| Secondary | 34 | 75.56% |
| Certificate | 8 | 17.78% |
| **Number of years of practice median (interquartile range), years** | 7 (3–15) | 100% |
| <6 | 17 | 38.6% |
| 6–10 | 12 | 27.3% |
| >10 | 16 | 35.6% |
| **Heard about mycetoma** | | |
| No | 36 | 80% |
| Yes | 9 | 20% |
| **Prior training on mycetoma** | | |
| No | 42 | 93.3% |
| Yes | 1 | 2.2% |
| I don't know | 2 | 4.4% |
| **Encountered an individual with suspected mycetoma in the last 6 months** | | |
| No | 30 | 66.7% |
| Yes | 14 | 31.1% |
| I don't know | 1 | 2.2% |

## Suspected cases of mycetoma encountered by community health workers six months before the training

Six months before the survey, 14 (31.8%) CHWs encountered 14 suspected cases of mycetoma. The majority (n = 13, 92.9%) involved men, and 12 (85.7%) cases affected the foot, either alone or with involvement of the leg and knee. While the majority of the CHWs (n = 11, 78.6%) referred the affected individuals to a health facility, 4 CHWs offered alternative advice, specifically the application of herbs (n = 2, 14.3%) or took no action (n = 2, 14.3%), (**Table 2).**

## Community health worker's knowledge about mycetoma

At baseline, the mean knowledge score was 7±2.42, and this improved to 11±1.99 in the immediate post-test score (p<0.001), and to 10±2.35 6 months post-training (p<0.001), without any additional training thereafter, **Fig 1**.

## Knowledge about mycetoma among community-health workers across different demographic groups

Compared to baseline scores, female participants had a statistically significant improvement in knowledge scores about mycetoma (5.4 (3.23) versus 11.2 (2.62), p = 0.004). The same trend was observed with participants aged 30–40 years (6.7 (2.65) versus 9.7 (1.82), p = 0.031), and those 40 years or older (6.8(2.18) versus 9.8(1.82), p = 0.035). Participants with a secondary

**Table 2. Details of suspected cases of mycetoma encountered six months before the training.**

| Variable | Freq (%) |
|---|---|
| **Sex** | |
| Female | 1(7.1%) |
| Male | 13(92.9%) |
| **Age median (interquartile range)** | 54(50–58) |
| <50 | 3(23.1%) |
| 50–60 | 8(57.14%) |
| >60 | 3(23.2%) |
| **Part of the body affected** | |
| Foot | 12 (85.7%) |
| Leg and knee | 1 (7.1%) |
| Hand | 1 (7.1%) |
| **Action taken** | |
| Referred to health facility | 11 (78.6%) |
| Advised patient to apply herbs | 2 (14.3%) |
| Nothing | 1 (7.1%) |

level of education also had a significant improvement in knowledge scores compared to those with lower levels of education (7.3(2.39) versus 9.9(2.51), p = 0.007), Table 3.

## Presumptive cases of mycetoma identified by CHWs after the training

Over the six months' period, the trained CHWs screened 2773 people in their respective communities and identified 30 presumptive cases of mycetoma with a median age of 30years, for which all had feet affected Table 4. **Fig 2** provides an image of one of these presumptive cases, showing multiple sinuses as identified by a CHW.

Image of one of the presumptive cases of mycetoma with multiple sinuses identified by a CHW

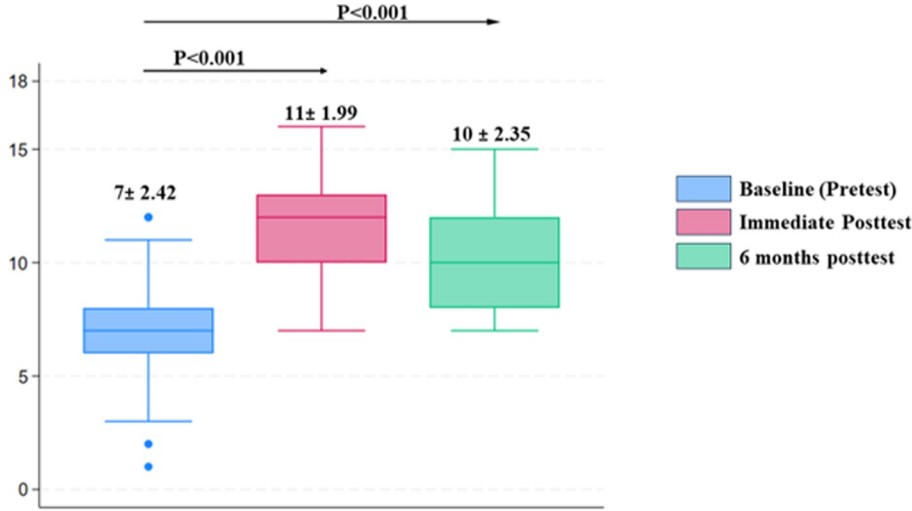

**Fig 1. Knowledge scores of community healthcare workers at baseline (before training), immediately after and 6 months after training.**

**Table 3. Mean knowledge scores.**

| Variable | Baseline (Pre-Test) | 6 months (Post-Test) | P-value |
|---|---|---|---|
| **Sex** | | | |
| Female | 5.4 (3.23) | 11.2 (2.62) | **0.004** |
| Male | 8.3 (2.14) | 9.6 (2.03) | 0.060 |
| **Age** | | | |
| <30 | 7.7(2.42) | 14.0(1.00) | 0.298 |
| 30–40 | 6.7(2.65) | 9.7(1.82) | **0.031** |
| >40 | 6.8(2.18) | 9.8(1.82) | **0.035** |
| **Highest level of education** | | | |
| Primary | 6.0(1.7) | 11.0(-) | - |
| Secondary | 7.3(2.39) | 9.9(2.51) | **0.007** |
| Certificate | 6.6(3.26.) | 10.5(4.95) | 0.083 |
| **Number of years of practice median (interquartile range)** | | | |
| <6 | 6.8(2.04) | 9.3(1.7) | 0.157 |
| 6–10 | 7.3(2.42) | 11.0(2.98) | 0.080 |
| >10 | 7.2(2.98) | 10.2(2.21) | 0.025 |

## Qualitative

Overall, 45 CHWs were interviewed through four focus group discussions with approximately 10–15 CHWs. Their socio-demographic characteristics are represented in Table 1.

## Theme one: experiences with mycetoma cases in the communities

Interaction with the CHWs revealed a considerable number of undiagnosed, misdiagnosed, or undetected mycetoma cases within the community. However, due to factors such as limited awareness among community members and the absence of a distinct indigenous name for the disease, mycetoma was often attributed to superstitious beliefs such as witchcraft, thus hindering timely diagnosis and intervention.

> *I have witnessed this disease in a nearby village. My brother has it, but no one in my village knows what it is called. It all started with swelling, which later developed into something resembling Goosebumps . . . the condition remains unchanged, with persistent bleeding wounds. (10 years' experience of CHW work, Gulu)*

**Table 4. Number of presumptive cases of mycetoma identified by CHWs after the training.**

| District | Gulu | | Pader | | Total |
|---|---|---|---|---|---|
| Follow-up at; | 3 months | 6 months | 3months | 6months | |
| Total number of people screened | 482 | 563 | 966 | 762 | 2773 |
| Females screened | 233 | 382 | 548 | 410 | 1573 |
| Males screened | 249 | 181 | 418 | 352 | 1200 |
| Females with presumptive cases | 3 | 0 | 2 | 1 | 6 |
| Males with presumptive cases | 7 | 4 | 9 | 4 | 24 |
| Total number of presumptive cases of mycetoma | 10 | 4 | 11 | 5 | 30 |

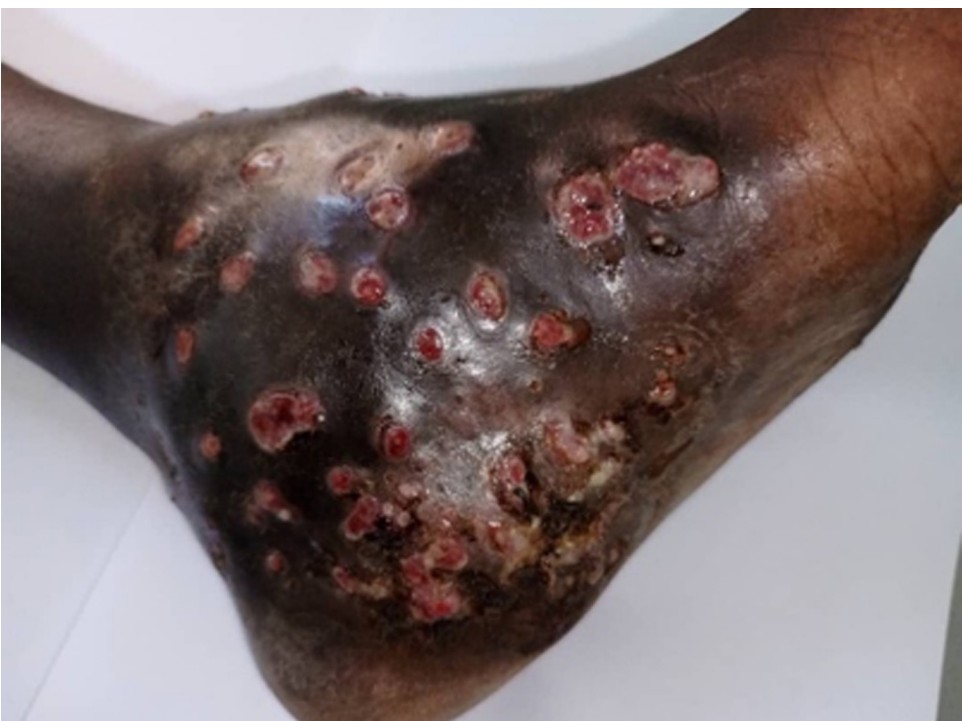

**Fig 2. A presumptive case of mycetoma identified by the community health care workers.**

> *I am aware of the presence of mycetoma. As I speak, it has affected my own family, particularly, my father. We sought medical care at Lacor hospital, but regrettably, there has been no improvement to date (15 years' experience of CHW work, Pader)*

## Challenges in identification

Limited knowledge and awareness of mycetoma, coupled with a restricted perception of its severity, contributed to delays in healthcare-seeking. Lack of understanding about the illness within local communities led to delayed medical attention as patients often endured symptoms until they became severe.

> *In my community, there are people who do not take this disease seriously. When it initially manifests, they often opt to seek care from traditional healers. . .However, as the sickness escalates, they eventually opt for proper health care having wasted so much money. (34 years, 8 years' experience of CHW work, Gulu)*

**Poor communication.**   Inadequate communication between the health workers and patients regarding the various aspects of the illness presented a significant barrier to treatment completion.

> As a caregiver for months, *I endured degrading treatment for health workers, akin to how one might treat a dog. Despite continuous inquiries, no explanations about my patient's*

*diagnosis or its cause were provided. Eventually, left with no choice, we left hospital. (38 years, 9 years' experience of CHW work Gulu)*

## Theme two: Knowledge and Awareness of Mycetoma

While a few CHWs possessed some knowledge about mycetoma symptoms, many lacked a comprehensive understanding of the scientific pathophysiology. Misidentification of mycetoma as other diseases, such as elephantiasis, was common among CHWs, leading to confusion in symptom explanation.

*The signs typically start as an initial itching sensation on the foot, which subsequently progresses to form blisters. These blisters turn into multiple hollow wounds. (29 years 3 years' experience of CHW work Gulu)*

## Theme three: Barriers to Early Diagnosis and Referral

**Stigma.** Stigma and discrimination emerged as significant barriers, with individuals fearing judgement and discrimination by others in the community. This fear often led to delayed healthcare seeking resulting in severe disease and increased morbidity.

*"I referred a woman with a similar leg problem, but she chose not to go to the health facility due to fear of rejection and discrimination, even in public transportation. Consequently, she opted to stay at home and self-treat with local herbs." (39 years, 5 years' experience of VHT work Gulu)*

**Influence in decision-making.** The decision-making process in seeking healthcare, especially for women, was influenced by delayed approval from their partners, from whom they had to seek permission. This delay significantly impacted the execution of referrals and access to care.

*In most cases, women who are suffering from mycetoma... do not go to the healthcare facility because they can't go and seek medical care without their partner's permission (32 years, 4 years' experience of CHW work Gulu)*

**Distance.** The absence of accessible transportation coupled with logistical constraints hindered timely access to health care providers, even following referral.

*The greatest challenge here is distance. Some villages are not easily accessible even to CHWs for health outreach programs due to long distance...community members find it had to access the health facility for medical help (34 years, 5 years' experience of CHW work Pader)*

**Alternative solutions.** Community members sought alternatives to the conventional health center systems such as traditional healers and private health facilities, causing delays in seeking standard care.

*Often when the distance to the government health facility is the problem, patients always look for alternative private health facilities like clinics, drug shops, and traditional healers to get medical care. (31 years, 7 years' experience of CHW work Gulu)*

### Theme four: Strategies to overcome barriers

**4.1 Capacity building among community health workers.** Capacity building through training of CHWs on the identification of mycetoma was reported as one of the possible solutions to the barriers to early diagnosis and referral of presumptive mycetoma cases in order to prevent associated morbidity.

*Initiatives such as educating CHWs on the use of clinical signs to identify suspected mycetoma cases will be able to help identify such cases early in the community so that we can refer them to the health centre for the appropriate medical care. (31 years, 7 years' experience of CHW work Gulu)*

**4.2. Collective Financial support.** Community and family financial support emerged as crucial players for overcoming financial barriers to accessing healthcare through collective initiatives such as fundraising and family assistance aid in covering medical costs.

*In the community, families or clans have resorted to collecting money to support the family. . .they have a collective clan saving to support families with health and other financial related issues (31 years, 7 years' experience of CHW work Gulu)*

### 5.0 Theme five: Training and Support on mycetoma diagnosis and referral

In the districts of Pader and Gulu, none of the CHWs had ever received training on mycetoma diagnosis and referral. Without training, CHWs lack the necessary knowledge and skills to recognize mycetoma symptoms and refer affected individuals to appropriate healthcare facilities for further evaluation and treatment.

*"We have never received any training on identifying or referring mycetoma cases. It is difficult for us to differentiate mycetoma from other skin conditions, and we often refer cases to general healthcare facilities without proper explanation to the affected individuals. As a result, some defer going to the hospital" (23 years, 3 years' experience of CHW work, Gulu).*

### Discussion

In this study, we aimed to identify knowledge gaps and evaluate the impact of training CHWs on mycetoma case detection in Northern Uganda. We found an inadequacy in knowledge on mycetoma among CHWs at baseline whose mean score was 7 out of 18. Notably, only a 20% and 31.8% of CHWs had prior awareness about mycetoma and had ever encountered a suspected case of mycetoma in the last six months prior to the training respectively. Furthermore, this study demonstrated a significant increase in knowledge scores, rising to an average of 11 immediately after the training and maintaining a score of 10 at the six months follow-up without additional training. These results suggest that the training was effective in both imparting

knowledge and sustaining it over time. Particularly significant improvements were observed among female CHWs, those aged 30-40years, and those with a secondary level of education.

Despite the fact that no prior studies assessing impact of training of CHWs on mycetoma have been done, several publications indicate that training CHWs in health subjects of public health concern significantly improves knowledge on the given subject. For instance, recent systematic review on the effectiveness of CHWs training programmes for cardiovascular disease management in LMICs demonstrated improved knowledge level post training and significant knowledge retention six months after the intervention [22]. Another study conducted in South Africa where CHWs were trained on national government priorities namely; HIV/AIDS, sexually transmitted disease, tuberculosis and women's sexual and reproductive health and rights showed marked knowledge improvements with eventual increase in confidence in advising clients and improved field based performance [23]. In Rwanda, Nigeria, South Africa, and Brazil, training of CHWs significantly improved knowledge, attitudes and practices on epilepsy, rheumatic heart disease, mental health and maternal and infant health respectively among CHWs [24–26]. As such, results from these studies align with findings from our study collectively highlighting the broader applicability and potential impact of well-structured training programs for CHWs across different health domains.

The ability of CHWs to identify 30 presumptive mycetoma cases over a six-month period further emphasize their potential role in community-based surveillance. The observed predominance of male cases and the frequent involvement of the feet align with global epidemiological patterns, which associate mycetoma with environmental and occupational factors prevalent in low-income settings [27–29]. Individuals living in low socioeconomic, warm, and humid environments are particularly at risk, as they are more likely to walk barefoot and engage in agricultural activities, increasing their susceptibility to traumatic inoculation of mycetoma-causing agents into the skin [7,30,31]. A retrospective study by Bonifaz and colleagues in Mexico reported a male predominance with a sex ratio of 3:1, predominantly affecting adults with a mean age of 34.5 years, and 62% of cases involving feet [32].

The qualitative findings from our study corroborated the quantitative findings with several factors such as limited knowledge and awareness, stigma and alternative healthcare-seeking behaviors emerging as barriers to early detection of mycetoma in Northern Uganda. In addition, the absence of an indigenous name for mycetoma further impeded the awareness of mycetoma and furthered its attribution to superstitions such as witchcraft. These myths and misconceptions further attest to how much mycetoma is a neglected disease in Uganda. Furthermore, our study identified several systemic barriers that hinder early identification and referral of mycetoma cases in the communities including delayed decision making processes and logistical constraints such as inaccessible transportation coupled with long distances to the health facilities impeding timely healthcare-seeking within affected communities, exacerbating the disease's progression and increasing morbidity [33].

Whereas the CHWs had not been previously engaged in the surveillance of mycetoma cases in Uganda, our findings indicate the willingness of CHWs to partake of community based initiatives to educate them and empower them in the identification and referral of mycetoma cases. The CHWs' receptiveness to the training and enthusiasm to disseminate knowledge among community members further highlight the need to leverage CHWs as key players in the mycetoma surveillance to improve awareness, dispel myths, reduce stigma, and control of mycetoma in Uganda.

The strengths of our study include the mixed-methods approach utilized in this study that enabled the exploration of the multifaceted perspectives of knowledge as well as barriers and challenges associated with mycetoma detection. Despite the fact that the study was a multicentre study, the small sample size limits the generalizability to the Ugandan population. We

acknowledge the potential recall bias among the participants due to the self-reported data regarding prior awareness of mycetoma and encounters with suspected cases affecting the reliability of the data. While efforts were made to minimize this bias through using structured data collection tools and interviewer training, the possibility of overestimation or underestimation of past experiences cannot be entirely ruled out. In addition, in this study, we did not follow-up with health facilities to determine the outcomes of the 30 suspected mycetoma cases referred by CHWs, which limits our understanding of the effectiveness of the referral process. This was also the first study to assess the impact of training CHWs on the knowledge and early detection and referral of mycetoma in Uganda.

## Conclusion

This study shows a significant knowledge a gap among CHWs regarding mycetoma, and the substantial improvement in knowledge following training. Furthermore, the identification of presumptive mycetoma cases by CHWs reflects the potential of leveraging their role in community-based surveillance and referral systems in Uganda. Therefore, further utilization CHW-led initiatives for mycetoma surveillance through capacity building is crucial in combating mycetoma and improving overall health outcomes in affected communities in Uganda.

## Supporting information

**S1 Data. Data set.**
(XLSX)

**S2 Data. STROBE checklist.**
(DOCX)

**S3 Data. COREQ checklist.**
(DOCX)

## Acknowledgments

We acknowledge and thank Dr. Kenneth Cana, District Health Officer of Gulu district, and Dr. Benson Oyoo, District Health Officer of Pader district for their valuable support during the study, Brenda Nakitto for her assistance with data analysis, and Fiona Gladys Laker for conducting the Focus group discussions.

 **Disclaimer**

 **Patient and public involvement**. Participants and the public will be informed of the conduct and findings of this study through the Ministry of Health and the District Health Offices.

 **Patient consent for publication.** Written informed consent will be obtained from all study participants for their data to be published.

## Author Contributions

**Conceptualization:** Winnie Kibone, Andrew Weil Semulimi, Richard Kwizera, Felix Bongomin.

**Data curation:** Winnie Kibone, Andrew Weil Semulimi, Richard Kwizera, Felix Bongomin.

**Formal analysis:** Winnie Kibone, Andrew Weil Semulimi, Felix Bongomin.

**Funding acquisition:** Winnie Kibone, Felix Bongomin.

**Investigation:** Winnie Kibone, Andrew Weil Semulimi, Richard Kwizera, Felix Bongomin.

**Methodology:** Winnie Kibone, Andrew Weil Semulimi, Richard Kwizera, Felix Bongomin.

**Project administration:** Winnie Kibone, Felix Bongomin.

**Resources:** Winnie Kibone, Andrew Weil Semulimi, Felix Bongomin.

**Software:** Winnie Kibone, Andrew Weil Semulimi, Richard Kwizera, Felix Bongomin.

**Supervision:** Winnie Kibone, Andrew Weil Semulimi, Richard Kwizera, Felix Bongomin.

**Validation:** Winnie Kibone, Andrew Weil Semulimi, Richard Kwizera, Felix Bongomin.

**Visualization:** Winnie Kibone, Andrew Weil Semulimi, Felix Bongomin.

**Writing – original draft:** Winnie Kibone, Felix Bongomin.

**Writing – review & editing:** Winnie Kibone, Andrew Weil Semulimi, Richard Kwizera, Felix Bongomin.

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
