## [Decision Letter · Decision Letter 0]

25 Jun 2024

Dear Dr Kibone,

Thank you very much for submitting your manuscript "Community-Based Mycetoma Surveillance in Uganda: Identifying Knowledge Gaps and Training of Community Health Workers to Improve Case Detection" for consideration at PLOS Neglected Tropical Diseases. As with all papers reviewed by the journal, your manuscript was reviewed by members of the editorial board and by several independent reviewers. The reviewers appreciated the attention to an important topic. Based on the reviews, we are likely to accept this manuscript for publication, providing that you modify the manuscript according to the review recommendations. 

Sincerely,

Joshua Nosanchuk, MD

Section Editor

Joshua Nosanchuk

Section Editor

Reviewer's Responses to Questions

**Key Review Criteria Required for Acceptance?**

**Methods**

-Are the objectives of the study clearly articulated with a clear testable hypothesis stated?

-Is the study design appropriate to address the stated objectives?

-Is the population clearly described and appropriate for the hypothesis being tested?

-Is the sample size sufficient to ensure adequate power to address the hypothesis being tested?

-Were correct statistical analysis used to support conclusions?

-Are there concerns about ethical or regulatory requirements being met?

Reviewer #1: -Are the objectives of the study clearly articulated with a clear testable hypothesis stated? yes

-Is the study design appropriate to address the stated objectives? yes

-Is the population clearly described and appropriate for the hypothesis being tested? yes 

-Is the sample size sufficient to ensure adequate power to address the hypothesis being tested? yes 

-Were correct statistical analysis used to support conclusions? yes 

-Are there concerns about ethical or regulatory requirements being met? none

Reviewer #2: Methods is clear

Reviewer #3: -The objectives of the study is clearly articulated with a clear testable hypothesis. 

-The study design is appropriate to address the stated objectives.

- Study population is clearly described and appropriate

- The sample size selection needs some statistical formula that is sufficient to ensure adequate power to address the hypothesis

-Acceptable statistical analysis was used to support conclusions.

**Results**

-Does the analysis presented match the analysis plan?

-Are the results clearly and completely presented?

-Are the figures (Tables, Images) of sufficient quality for clarity?

Reviewer #1: Additional statistical analysis could be performed on the collected data to enhance the study’s findings.

 Photos on the methodology will be helpful

Reviewer #2: the result presented in a good manner

re-edite the tables

Reviewer #3: -The analysis presented did match the analysis plan.

-These results are clearly and completely presented.

-Figures (Tables, Images) are of sufficient quality for clarity.

**Conclusions**

-Are the conclusions supported by the data presented?

-Are the limitations of analysis clearly described?

-Do the authors discuss how these data can be helpful to advance our understanding of the topic under study?

-Is public health relevance addressed?

Reviewer #1: -Are the conclusions supported by the data presented? yes 

-Are the limitations of analysis clearly described? study done in limited area 

-Do the authors discuss how these data can be helpful to advance our understanding of the topic under study? yes 

-Is public health relevance addressed? yes

Reviewer #2: Yes

Reviewer #3: -The conclusions are supported by the data presented.

-Limitations of analysis needs to clearly,

-Discussion of these data would be helpful to advance the understanding of the topic under study?

-Public health relevance is clearly addressed.

**Editorial and Data Presentation Modifications?**

Reviewer #1: (No Response)

Reviewer #2: (No Response)

Reviewer #3: Editorial suggestions/or relatively minor modifications are suggested earlier for sample size.

**Summary and General Comments**

Reviewer #1: The research concept is robust, addressing the crucial issue of early detection of mycetoma to improve management 

 and outcomes in remote rural communities. The study examines the knowledge of community health workers on the f 

 front lines, assessing their ability to identify this neglected disease.

 Please consider the following points: 

 • It should be explicitly stated in the manuscript that these workers do not aim to diagnose mycetoma but rather to 

 detect suspected cases for referral to appropriate clinical, laboratory, and imaging diagnostic facilities.

 • Out of the 300 suspected patients, it is essential to clarify their subsequent outcomes, especially from an ethical 

 standpoint.

 • If confirmed as mycetoma, this cohort would significantly contribute to understanding the prevalence of the disease in 

 Uganda.

 • The objectives and methodology of the study were clearly articulated.

 • Ethical considerations were thoroughly addressed.

 • Additional statistical analysis could be performed on the collected data to enhance the study’s findings.

 • The discussion section is relatively weak, with numerous repetitions from the results section.

 • Photos on the methodology will be helpful

 • Overall, the manuscript requires substantial revision to improve the quality of English and correct typographical 

 errors.

 • I have highlighted several statements in yellow with comments directly on the manuscript.

 Thank you for considering these suggestions. I believe addressing these points will greatly enhance the quality and 

 impact of the study.

Reviewer #2: (No Response)

Reviewer #3: See Attachment

PLOS authors have the option to publish the peer review history of their article (what does this mean?). If published, this will include your full peer review and any attached files.

Reviewer #1: Yes: Prof Ahmed Fahal

Reviewer #2: Yes: MOHAMED DAFFALLA AWADALLA GISMALLA

Reviewer #3: Yes: Mohamed Zain Ali

Figure Files:

Data Requirements:

Reproducibility:

References

---

## [Decision Letter · Decision Letter 1]

24 Sep 2024

Dear Dr Kibone,

Thank you for your thoughtful and complete revision of this work. We are pleased to inform you that your manuscript 'Community-Based Mycetoma Surveillance in Uganda: Identifying Knowledge Gaps and Training of Community Health Workers to Improve Case Detection' has been provisionally accepted for publication in PLOS Neglected Tropical Diseases.

Best regards,

Joshua Nosanchuk, MD

Section Editor

Joshua Nosanchuk

Section Editor

Reviewer's Responses to Questions

**Key Review Criteria Required for Acceptance?**

**Methods**

-Are the objectives of the study clearly articulated with a clear testable hypothesis stated?

-Is the study design appropriate to address the stated objectives?

-Is the population clearly described and appropriate for the hypothesis being tested?

-Is the sample size sufficient to ensure adequate power to address the hypothesis being tested?

-Were correct statistical analysis used to support conclusions?

-Are there concerns about ethical or regulatory requirements being met?

Reviewer #1: (No Response)

**Results**

-Does the analysis presented match the analysis plan?

-Are the results clearly and completely presented?

-Are the figures (Tables, Images) of sufficient quality for clarity?

Reviewer #1: (No Response)

**Conclusions**

-Are the conclusions supported by the data presented?

-Are the limitations of analysis clearly described?

-Do the authors discuss how these data can be helpful to advance our understanding of the topic under study?

-Is public health relevance addressed?

Reviewer #1: (No Response)

**Editorial and Data Presentation Modifications?**

Reviewer #1: (No Response)

**Summary and General Comments**

Reviewer #1: (No Response)

PLOS authors have the option to publish the peer review history of their article (what does this mean?). If published, this will include your full peer review and any attached files.

Reviewer #1: **Yes: **Prof Fahal

---

## [Editor Report · Acceptance letter]

1 Oct 2024

Dear Dr Kibone,

We are delighted to inform you that your manuscript, "Community-Based Mycetoma Surveillance in Uganda: Identifying Knowledge Gaps and Training of Community Health Workers to Improve Case Detection," has been formally accepted for publication in PLOS Neglected Tropical Diseases.

Best regards,

Shaden Kamhawi

co-Editor-in-Chief

Paul Brindley

co-Editor-in-Chief
